# Rhizosphere Microbe Affects Soil Available Nitrogen and Its Implication for the Ecological Adaptability and Rapid Growth of *Dendrocalamus sinicus*, the Strongest Bamboo in the World

**DOI:** 10.3390/ijms241914665

**Published:** 2023-09-28

**Authors:** Peitong Dou, Qian Cheng, Ning Liang, Changyan Bao, Zhiming Zhang, Lingna Chen, Hanqi Yang

**Affiliations:** 1Institute of Highland Forest Science, Chinese Academy of Forestry, Kunming 650233, China; lkyzksdpt@163.com (P.D.); cqq08042@163.com (Q.C.); ln417@126.com (N.L.); baochangyan929@163.com (C.B.); zhangzhiming0406@163.com (Z.Z.); 2College of Life Science, Xinjiang Normal University, Xinyi Road, Shayibake District, Urumqi 830054, China; 3Key Laboratory of Breeding and Utilization of Resource Insects, National Forestry and Grassland Administration, Kunming 650233, China

**Keywords:** *Dendrocalamus sinicus*, ecological adaption, microbial function, red soil, rhizosphere microbes, rapid growth

## Abstract

The interaction between soil microbes and plants has a significant effect on soil microbial structure and function, as well as plant adaptability. However, the effect of soil micro-organisms on ecological adaption and rapid growth of woody bamboos remains unclear. Here, 16S rRNA and ITS rRNA genes of rhizosphere micro-organisms were sequenced, and the soil properties of three different types of *Dendrocalamus sinicus* were determined at the dormancy and germination stages of rhizome buds. The result showed that each type of *D. sinicus* preferred to absorb ammonia nitrogen (NH_4_^+^-N) rather than nitrate nitrogen (NO_3_^−^-N) and required more NH_4_^+^-N at germination or rapid growth period than during the dormancy period. In total, nitrogen fixation capacity of soil bacteria in the straight type was significantly higher than that in the introduced straight type, while the ureolysis capacity had an opposite trend. Saprophytic fungi were the dominant fungal functional taxa in habitat soils of both straight and introduced straight type. Our findings are of great significance in understanding how soil microbes affect growth and adaptation of woody bamboos, but also for soil management of bamboo forests in red soil.

## 1. Introduction

Red soil is an important soil type in southern China. Due to soil parent material, high temperature, high humidity and other climatic factors, red soil is generally characterized by high Fe contents, low pH, low contents of organic carbon and soil nutrient elements, as well as strong erosion [1,2]. Nitrogen (N) is one of the major elements essential for plant growth [3]. Ammonia nitrogen (NH_4_^+^-N) and nitrate nitrogen (NO_3_^−^-N) are the main forms of N uptake by plants, originating primarily from microbial processes of ammonification and nitrification [4,5]. Noticeably, soil N availability in terrestrial ecosystems commonly limits plant growth [6] and is influenced by the interaction between plants and rhizosphere microbes [5]. Rhizosphere is the interface between plant roots and soil [7]. As an important part of the rhizosphere system, rhizosphere micro-organisms are considered as an important part of the extended plant phenotypes, which could greatly promote the interaction between plants and the environment [8]. In general, plants recruit micro-organisms to the rhizosphere by releasing secretions (such as sugars, organic acids) [5], which will increase rhizosphere microbial abundance and activity and affect its community structure and function [9]. Conversely, rhizosphere micro-organisms also help plants absorb nutrients from the soil [10], improve plant growth and development [11], protect host plants from pathogens [12] and increase their tolerance to abiotic stresses [13,14]. Meanwhile, a growing number of studies also indicate that spatial variation in soil microbe communities may affect local adaptation patterns of plants [8,15,16]. Abiotic environmental variation, such as soil nutrients and aridity, would interact with microbe communities and affect plant fitness [17,18]. In summary, interaction among soil, plants and soil microbes plays a critical role in plant nutrition acquisition, healthy growth and geographical range margins [19,20,21,22].

Woody bamboo belongs to the bamboo subfamily of *Gramineae* (*Poaceae*), with ca. 80 genera and more than 1500 species [23]. Yunnan Province of China is a world-renowned laterite plateau and possesses more than 220 native woody bamboos from 28 genera occurring in the red soil [24]. Considering that woody bamboos are of crucial economic, ecological and cultural value in their distribution area [24], the introduction and cultivation of excellent bamboo resources have attracted increasing interest in both academia and industry in recent years [25]. The rapid growth of young culms is one of the most striking biological features of woody bamboos [26,27]. Due to low contents of organic carbon and soil nutrient in acidic red soil [1], rapidly growing bamboo plants may be more susceptible to the influence of rhizosphere micro-organisms on their nutrient absorption [11]. Furthermore, as one of the most important members of the bamboo forest ecosystem, the structure and function of soil microbial communities in bamboo rhizosphere have also received increasing attention from scientists. Recently, it was reported that rhizosphere micro-organisms were involved in carbohydrate degradation and nitrogen fixation process of woody bamboos rhizomes, which contributed to the annual shooting of *Cephalostachyum pingbianense* [11]. The bacterial genera *Flavobacterium*, *Bacillus* and *Stenotrophomonas* might affect the flowering time of Moso bamboo (*Phyllostachys edulis*) by regulating the effective utilization of nitrogen [28]. Supplementing azotobacter and vesicular-arbuscular mycorrhizae fungi could increase the biomass of *Bambusa vulgaris* [29]. These studies suggested that soil micro-organisms can affect the growth, phenology and reproduction characteristics of woody bamboos. However, the effect of micro-ecological factors, especially soil micro-organisms, on ecological adaption and rapid growth of woody bamboos remains unclear.

Remarkably, *Dendrocalamus sinicus*, endemic to southern and southwestern Yunnan, China, is the strongest bamboo, which has been recorded in the world, with a diameter at breast height (DBH) of 30 cm and a height of nearly 30 m. The average fresh weight of the culm is 100–150 kg, and the yield of culm timber per unit area was five to eight times higher than that of *Phyllostachys edulis*, the most important economic bamboo species in the world [30]. These excellent characteristics make *D. sinicus* one of the most promising economic bamboo species in southern China. In nature, *D. sinicus* has two main stable culm-shape variants: the straight type and bending type [31]. They thrive in lateritic soil and inhabit environments, which share notable similarities. However, the narrow natural distribution of *D. sinicus* poses a limitation to its cultivation and utilization [25]. In the previous studies, we investigated its biological characteristics, including the mating system [32] and culm development [26,31]. Furthermore, our study investigated the impact of climate factors on the distribution of *D. sinicus*, revealing a substantially broader potential distribution range compared to its current range [25]. To identify the potential contribution of the rhizosphere microbes in the spatial distribution and growth of *D. sinicus*, we performed high-throughput sequencing of ITS rRNA and 16S rRNA genes from soil samples, determined some key soil properties and investigated the DBH and bamboo shooting ratio (BSR) in different types of *D. sinicus* (Figure 1). This study had two aims: (1) to explore the influence of micro-ecological factors on the success of woody bamboos introduction; (2) to determine the nitrogen absorption characteristics of *D. sinicus* and identify the potential microbial taxa, which are conducive to its rapid growth. 

## 2. Results

### 2.1. Soil Properties and Biological Characteristics of Various D. sinicus at Different Stages

The soil properties of different *D. sinicus* types were analyzed (Table 1). In May, the soil pH was significantly higher for each *D. sinicus* type compared to August (*p* < 0.05). The introduced straight type exhibited significantly higher soil organic carbon (SOC) content in May compared to August (*p* < 0.05). In each period, the SOC content ranked highest in the straight type, followed by the bending type and then the introduced straight type, with significant differences observed (*p* < 0.05). There was no significant difference in the soil AP content for the bending type between May and August (*p* > 0.05). However, the soil AP content for both the straight and introduced straight types was significantly higher in May compared to August (*p* < 0.05). In each period, the straight type had the highest soil available phosphorus (AP) content, followed by the introduced straight type and then the bending type, with significant differences observed (*p* < 0.05). The introduced straight type had significantly lower soil available potassium (AK) content compared to the bending and straight types in each stage (*p* < 0.05), suggesting a higher AK absorption by *D. sinicus* in native areas. In May, the soil NO_3_^−^-N content for the bending type was significantly higher than that in August (*p* < 0.05) and consistently highest at two periods (*p* < 0.05). Conversely, the soil NH_4_^+^-N content for all types was significantly lower in May compared to August (*p* < 0.05), indicating increased NH_4_^+^-N absorption during shooting and rapid growth period. The introduced straight type had significantly lower soil water content (SWC) compared to the bending and straight types in each period (*p* < 0.05). Apart from the straight type in August, which exhibited significantly higher SWC than that in May (*p* < 0.05), there were no significant differences in SWC between May and August for other types (*p* > 0.05). There was no significant difference in the C: N among different types during the same period (*p* > 0.05). However, the NH_4_^+^: NO_3_^−^ of the straight type was significantly higher than that of the bending type and the introduced straight type during the same period (*p* < 0.05). The biological characteristics of various *D. sinicus* were compared as follows: the straight type had the largest DBH and bamboo stem weight (BSW), followed by the bending type and then the introduced straight type, with significant differences observed (*p* < 0.05). The introduced straight type showed a significantly lower BSR compared to both the bending type and the straight type. 

### 2.2. The α Diversity and β Diversity of Soil Microbial Community

A total of 1,614,330 bacterial and 2,554,342 fungal raw reads were obtained from 30 soil samples. After the paired-end reads were spliced and filtered, clean tags for 1,383,699 bacteria and 2,307,941 fungi were obtained. These tags were classified into 2577 bacterial and 2565 fungal OTUs. One-way analysis of variance (ANOVA) was performed to compare the α diversity of microbial communities (bacterial and fungal) among the soil samples (Figure 2). The Shannon index values for bacteria in the rhizosphere soil of both bending and straight types were significantly higher in May compared to August (Figure 2C). In May, no significant difference in the α diversity of bacterial community was observed among the different soil samples (*p* > 0.05). However, in August, the Shannon index of bacterial community in the rhizosphere soil of the introduced straight type was significantly higher than in other types (*p* < 0.05). The fungal Sobs and Chao1 index values for the straight type were significantly lower in May compared to August (*p* < 0.05) (Figure 2B,F). In May, the fungal α diversity index in the rhizosphere soil of the bending type was significantly higher than that of the straight type (*p* < 0.05) (Figure 2B,D,F). However, in August, there was no significant difference in the fungal α diversity index among the different soil samples (*p* > 0.05).

The PCoA analysis demonstrated a clear differentiation of microbial communities in various rhizosphere soil samples. Specifically, the fungal PCoA (Figure 2H) exhibited a more distinct separation among samples compared to the bacterial PCoA (Figure 2G). Furthermore, the ANOSIM analysis indicated significant effects of the *D. sinicus* type and growth period on the structure of soil bacterial (r = 0.8441, *p* = 0.001) and fungal communities (r = 0.8646, *p* = 0.001) (Appendix A).

### 2.3. Soil Microbial Community Composition and Function

The microbial community structure and function in the rhizosphere soil varied among different types. Actinobacteria and Proteobacteria were the dominant bacterial phyla, together comprising over 50% of the bacterial communities in each soil sample (Figure 3A, Appendix A). The relative abundance of Actinobacteria peaked in August for the bending type (44.19%) and reached its lowest point in May for the straight type (22.99%) (Figure 3A, Appendix A). In contrast, the relative abundance of Proteobacteria peaked in May for the bending type (32.93%) and reached its lowest point in August for the introduced straight type (15.47%). The relative abundance of Actinobacteria in each type increased from May to August, while the relative abundance of Proteobacteria decreased during the same period. The dominant bacterial genera in all soil samples were *Bacillus*, *Bradyrhizobium*, *Mycobacterium*, *Kitasatospora*, *Sphingomonas*, *RB41*, *Gaiella* and *Haliangium* (Figure 3C, Appendix A). As for fungi, the dominant fungal phyla were Ascomycota and Basidiomycota, accounting for over 90% of the fungal communities in each soil sample (Figure 3B, Appendix A). The relative abundance of Ascomycota peaked in May for the bending type (68.57%) and reached its lowest point in May for the straight type (43.28%). Moreover, the relative abundance of Basidiomycota was highest in May for the straight type (52.09%) and lowest in May for the bending type (23.42%). The dominant fungal genera in all soil samples were *Cladophialophora*, *Trichoderma* and *Penicillium* (Figure 3D, Appendix A). In May, the rhizosphere soil of the bending type had significantly higher abundances of *Apiotrichum*, *Fusarium*, *Exophylla*, *Acremonium*, *Cordana* and *Neocosmospora* compared to the straight type (*p* < 0.05) (Appendix A). Conversely, *Tremelodendropsis*, *Archaeorhizogenes*, *Camarophyllus* and *Leohumicola* exhibited significantly higher abundances in the rhizosphere soil of the straight type compared to the bending type (*p* < 0.05). This result indicates that fungal communities beneficial to the growth of bending and straight types developed in their respective rhizospheres under natural conditions. 

Functional predictions of the bacterial community indicated that chemoheterotrophy, aerobic chemoheterotrophy, nitrogen fixation and nitrate reduction were the primary soil ecological functions among the rhizosphere soil samples (Figure 4A). This result suggested that the rhizosphere bacterial community of *D. sinicus* hada high capacity for biological nitrogen fixation. The abundance of cellulolysis in all soil samples was notably high in August, indicating a strong carbohydrate conversion ability of the rhizosphere bacterial community during the shooting and rapid growth period. Furthermore, the introduced straight type exhibited significantly higher ureolysis abundance compared to other samples in August (*p* < 0.05) (Figure 4A, Appendix A). The rhizosphere bacterial taxa were classified into nine phenotypes (Figure 4C), showing a wide variation in the relative abundance of different phenotypic bacteria within each sample. For instance, the relative abundance of aerobic bacteria was much higher than that of anaerobic bacteria. Notably, the relative abundance of potentially pathogenic bacteria in May was significantly higher in the straight type compared to the introduced straight type (*p* < 0.05) (Figure 4D). 

Figure 4B shows the functional composition of fungal communities in six soil samples and three types of trophic modes: saprotroph, symbiotroph and pathoproth. Saprotrophic fungi dominated the rhizosphere soil samples, constituting over 80.93% of the community (Appendix A). This group included various types, such as undefined saprotrophs, soil saprotrophs and wood saprotrophs. Pathogens comprised 4.39% to 12.46% of the community, mainly consisting of animal and plant pathogens. Symbiotrophs represented the smallest proportion, ranging from 1.22% to 14.69%. In particular, the straight type had a symbiotroph proportion of 14.69% in August (Appendix A), while the introduced straight type had a symbiotroph proportion of 3.72% in August.

### 2.4. Soil Microbial Taxa with Significant Differences 

LEfSe analysis was used to identify microbial taxa with significant abundance differences among the 30 most abundant genera in soil samples (Figure 5). This analysis revealed 26 taxonomic clades, which displayed differential abundance as bacterial biomarkers (Figure 5A, Appendix A). Soil samples of the straight type in May and August exhibited the fewest bacterial taxa with significant differences, consisting of two genera each. In May, the two genera were *Candidatus_Solibacter* and *Anaeromyxobacter*, while in August, *Bacillus* and *Bradyrhizobium* were identified. In contrast, soil samples of the bending type in May displayed the most bacterial taxa, including *Pedomicrobium*, *Solirubrobacter*, *Gaiella*, *Micromonospora*, *Arthrobacter*, *Nocardioides*, *Reyranella*, *RB41*, *MND1* and *Haliangium*. 

Meanwhile, a total of 20 fungal clades with significant abundance differences were identified in this study (Figure 5B, Appendix A). Specifically, soil samples of the straight type in August displayed the fewest fungal taxa, with only one genus, *Pseudallescheria*, showing notable variation. In contrast, soil samples of the bending type in May exhibited the most fungal taxa, including seven genera: *Gonytrichum*, *Acremonium*, *Neocosmospora*, *Mortierella*, *Apiotrichum*, *Exophiala* and *Fusarium*. 

### 2.5. Soil Properties Affecting the Diversity and Structure of Microbial Community 

The correlation analysis between soil properties and microbial α diversity indices revealed significant associations. Bacterial Sobs and Shannon index showed negative correlations with SWC and NH_4_^+^-N (*p* < 0.05), and bacterial Chao1 indices exhibited a negative correlation with SWC (*p* < 0.05) (Figure 6A). In fungal communities, the pH displayed a positive correlation with the Shannon index (*p* < 0.05), and soil AK and NH_4_^+^-N contents were positively correlated with Sobs and Chao1 indices (*p* < 0.05) (Figure 6B). Conversely, soil AP content was negatively correlated with all α diversity indices of fungi (*p* < 0.05). A positive correlation was observed between SWC and fungal Chao1 index (*p* < 0.05) (Figure 6B). RDA analysis revealed that the SOC, pH and NH_4_^+^-N were the primary soil properties affecting the structure of the bacterial community (r^2^ = 0.7487, *p* = 0.001 for SOC; r^2^ = 0.7109, *p* = 0.001 for pH; r^2^ = 0.6989, *p* = 0.001 for NH_4_^+^-N) (Figure 6C, Appendix A). Similarly, soil AK, SOC, AP and pH were identified as the main factors affecting the structure of the fungal community (r^2^ = 0.8171, *p* = 0.001 for AK; r^2^ = 0.8020, *p* = 0.001 for SOC; r^2^ = 0.7794, *p* = 0.001 for AP; r^2^ = 0.7727, *p* = 0.001 for pH) (Figure 6D, Appendix A). In particular, soil pH and SOC were found to affect the structure of both bacterial and fungal communities. 

### 2.6. Correlations of Microbial Genera with Soil Properties and Biological Characteristics of D. sinicus

The correlation analysis between the 30 most abundant microbial genera and soil properties revealed significant associations (Figure 7). In bacterial communities, *Reyranella*, *Paenibacillus*, *Micromonospora*, *Nocardioides*, *Luedemannella*, *Pedomicrobium*, *Bradyrhizobium* showed positive correlations with soil NH_4_^+^-N content (*p* < 0.05). Meanwhile, most of these genera showed positive correlations with SWC, SOC, DBH and BSW (*p* < 0.05) (Figure 7A). In the fungal community, *Penicillium*, *Entoloma*, *Hygrocybe*, *Tremellodendropsis*, *Archaeorhizomyces*, *Camarophyllus* and *Leohumicola* were positively correlated with soil AP content (*p* < 0.05). *Trichoderma*, *Mortierella*, *Trechispora*, *Apiotrichum*, *Sarcodon*, *Exophiala*, *Pseudallescheria*, *Cordana*, *Gonytrichum* and *Acremonium* showed positive correlations with soil AK content (*p* < 0.05). Moreover, *Mortierella*, *Trechispora*, *Apiotrichum*, *Exophiala*, *Tremellodendropsis* exhibited positive correlations with SWC, SOC, DBH and BSW (*p* < 0.05). *Trichoderma*, *Mortierella*, *Trechispora*, *Apiotrichum*, *Exophiala*, *Acremonium* and *Cordana* displayed positive correlations with soil NH_4_^+^-N content in fungal communities (Figure 7B). 

### 2.7. The Relationship between Selected Soil Properties and the Biological Characteristics of D. sinicus

The regression analysis revealed that soil NH_4_^+^-N content explained 52.43% of the variation in BSW (*p* < 0.05) (Figure 8B) and 63.44% of the variation in BSR (*p* < 0.05) (Figure 8F). Conversely, soil NO_3_^−^-N content explained only 7.05% of the variation in BSW (*p* < 0.05) (Figure 8A) and 61.27% of the variation in BSR (*p* < 0.05) (Figure 8E). Furthermore, the C: N and NH_4_^+^: NO_3_^−^ demonstrated a lower contribution to variation in BSW and BSR compared to the soil NH_4_^+^-N. The C: N and NH_4_^+^: NO_3_^−^ explained 10.82% and 31.04% of the variation in BSW (*p* < 0.05) (Figure 8C,D), as well as 23.73% and 34.94% of the variation in BSR (*p* < 0.05) (Figure 8G,H). This result indicated that the NH_4_^+^-N availability had a greater impact on the formation of biological traits of *D*. *sinicus*. 

## 3. Discussion

### 3.1. Characteristics of NH_4_^+^-N Absorption by D. sinicus

Soil nitrogen availability is a critical factor limiting forest land productivity [6,33]. NH_4_^+^-N and NO_3_^−^-N are the primary nitrogen forms absorbed by plants, with plant species exhibiting an absorption preference for NH_4_^+^-N or NO_3_^−^-N [6,34,35]. This study showed that each type of *D. sinicus* had an absorption preference for NH_4_^+^-N, as evidenced by the higher NH_4_^+^-N content in the rhizosphere soil compared to NO_3_^−^-N during both dormancy and shooting periods of the underground rhizome. The preference was likely attributed to distinct modes of absorption for NH_4_^+^-N and NO_3_^−^-N by roots, with NH_4_^+^-N uptake relying on ion exchange and NO_3_^−^-N uptake involving an active process [36]. NH_4_^+^-N could affect the effective number of carriers for absorption of NO_3_^−^-N [37], which may reduce NO_3_^−^-N absorption in plants. Meanwhile, plants exhibited a greater efficiency in converting NH_4_^+^-N into amides or amino acids compared to NO_3_^−^-N, and the energy requirement for NH_4_^+^-N absorption and assimilation was significantly lower than that of NO_3_^−^-N in plants [36]. Additionally, during the rainy season, NO_3_^−^-N was susceptible to leaching and runoff, leading to significant loss and transport of this nitrogen compound [11,35]. 

The uptake of NH_4_^+^-N and NO_3_^−^-N varies across different growth stages in plants [3]. In rice (*Oryza sativa* L.), NH_4_^+^-N played a dominant role during the vegetative growth stage, accounting for 68.9% of the total absorbed nitrogen. The absorption ratio of the two is around one during the reproductive growth period [3]. In this study, the NH_4_^+^: NO_3_^−^ of different *D. sinicus* types showed an increasing trend from May to August, aligning with the corresponding changes in soil NH_4_^+^-N content (Table 1). This result suggested that a higher NH_4_^+^: NO_3_^−^ was more crucial for the rapid growth of *D. sinicus* compared to the dormancy period of the underground rhizome. Moreover, a higher NH_4_^+^: NO_3_^−^ was found to enhance photosynthetic characteristics and promote root growth in bamboo plants [38,39]. The regression analysis also revealed that the NH_4_^+^: NO_3_^−^ explained 31.04% of the variation in BSW (*p* < 0.05), as well as 34.94% of the variation in BSR (*p* < 0.05) (Figure 8D,H), and soil NH_4_^+^-N content exhibited the highest contribution rate to both BSW and BSR of *D. sinicus*. Therefore, it was recommended to apply NH_4_^+^-N during the rapid growth period of *D. sinicus* in production practices. 

The soil C: N is an indicator of the equilibrium between soil carbon and nitrogen nutrition [40]. Typically, the soil C: N is around 25:1 [41]. In this study, the C: N in the rhizosphere soil varied between 11.39 and 15.34. A lower soil C: N enhanced the mineralization and release of N, which was available for plant uptake [42]. Soil parent material, soil type, topography and land use type are influential factors affecting the soil C: N, with minimal variation observed under similar conditions [40]. Therefore, the similar soil parent material and types in the habitats of various *D. sinicus* could account for the absence of a significant difference in soil C: N at the same growth stage. This finding provided valuable insights for selecting suitable planting locations for *D. sinicus*. 

### 3.2. Influence of Rhizosphere Microbes on NH_4_^+^-N Availability in Red Soil 

The natural distribution of *D. sinicus* is limited to the red soil habitat of Yunnan, China [30]. Red soil develops in warm temperate and humid climates and is characterized by low pH and soil nutrient availability [2]. In this study, the soil samples displayed a slightly acidic condition, with pH values ranging from 5.58 to 6.13. The RDA analysis confirmed that soil pH significantly affected both fungal (r^2^ = 0.7727, *p* = 0.001) and bacterial community structures (r^2^ = 0.7109, *p* = 0.001). This result showed that soil pH was a crucial factor influencing microbial community structure, which was consistent with previous studies [43]. Previous studies found that near-neutral pH soil facilitated the mineralization of carbon and nitrogen [44], providing a substrate for the microbial process [45,46]. Meanwhile, this study also showed that SOC had a significant impact on both bacterial (r^2^ = 0.7487, *p* = 0.001) and fungal communities (r^2^ = 0.8020, *p* = 0.001) (Appendix A). It was easy to understand that SOC was the core of the cycling and transformation of soil nutrient [47], and soil carbon decomposition was mediated by the microbial community [45]. To sum up, near-neutral pH and increased organic C availability were associated with increased microbial biomass [43,45]. 

Nitrogen (N) is one of the major elements essential for plant growth [6,48]. Soil N ammoniation and nitrification, as key processes in nitrogen cycling and plant nutrition absorption, are mainly mediated by soil microbes [4,5]. Currently, many studies found that microbial nitrogen fixation and mineralization were important ways of increasing the available nitrogen content in rhizosphere soil [10,35,48]. In the case of *Bambusa vulgaris*, supplementation of nitrogen fixing bacteria and vesicular-arbuscular mycorrhizae fungi could significantly increase growth [29]. Moreover, the soil NH_4_^+^-N content increased during the invasion of broadleaf forests by Moso bamboo, and the presence of the fungal lcc gene explained a significant portion of variations in net ammonification rate [35]. In this study, the prediction results of bacterial functions showed dominant roles of nitrogen fixation, cellulolysis and ureolysis in the rhizosphere soil of *D. sinicus*, indicating a high capacity for carbohydrate conversion and biological nitrogen fixation [11]. Remarkably, the rhizosphere microbes of the straight type possessed higher nitrogen fixing capacity, which could lead to higher soil NH_4_^+^-N content compared to the introduced straight type (Table 1, Appendix A) [48]. Furthermore, the FUNGuild analysis revealed a high proportion of saprotrophic fungi, which enhanced the availability of carbon and nitrogen [49,50]. 

### 3.3. Influence of Micro-Ecological Factors on the Adaptation of D. sinicus 

The rhizosphere micro-ecology of plants is influenced by the secretion of secondary metabolites, the taxa of rhizosphere microbes and the physicochemical properties of the soil. The disruption of micro-ecological balance can result in soil degradation, thereby influencing plant growth [51]. Previous studies indicated that plant adaptability was influenced by soil conditions and rhizosphere microbial taxa [52,53]. In this study, although the introduced straight type had the same origin region as the straight type, the introduced straight type exhibited significantly smaller DBH, BSW and BSR (*p* < 0.05) (Table 1). Analysis of soil properties revealed that the rhizosphere soil of the introduced straight type had significantly lower levels of SOC, AP, AK and NH_4_^+^-N compared to the straight type at each stage (*p* < 0.05). This result indirectly suggested that the introduced straight type had fewer available nutrients in the rhizosphere soil, which could lead to the decrease in DBH, BSW and BSR [54]. Meanwhile, apart from a significantly higher Shannon index of bacterial community observed in August for the introduced straight type compared to the straight type, no statistically significant differences were found in other indices between the two types (Figure 2). The prediction results of soil bacterial functions showed that nitrogen fixation abundance was significantly lower in the introduced straight type compared to the straight type at each stage (Figure 4A, Appendix A), which may be an important factor leading to the low soil NH_4_^+^-N content of the introduced straight type [48]. It was important to note that the rhizosphere soil of the introduced straight type exhibited a significantly higher abundance of the ureolysis function compared to the straight type in August (*p* < 0.05) (Figure 4A, Appendix A). This increase in the ureolysis function may be a response of micro-organisms to the NH_4_^+^-N demand for introduced straight type [55]. Additionally, the introduced straight type showed a significantly lower relative abundance of potentially pathogenic bacteria in May compared to the straight type (*p* < 0.05) (Figure 4D). These findings suggested that the introduction site had fewer pathogens and mutualistic bacteria (e.g., nitrogen fixation). The interaction among soil, plants and soil microbes was likely to influence the adaptation of the plant, as supported by previous research [19,20].

Similarly, the edaphic condition after introduction could affect the survival and growth of soil micro-organisms, force existing micro-organisms to adapt to the new environment and promote an increase in the abundance of specific micro-organisms [19]. The rhizosphere soil of both the straight and introduced straight type exhibited distinct fungal taxa according to the LEfSe analysis (Appendix A). The biomarkers for the straight type included *Leohumicola*, *Archaeorhizomyces*, *Pseudallescheria*, while the introduced straight type showed biomarkers such as *Cladophialophora*, *Entoloma*, *Agaricus*, *Clavaria* and *Chaetomium*. These microbial taxa were known to play roles in decomposing organic matter and increasing nitrogen and phosphorus availability [56,57]. These results highlighted the role of micro-organisms in increasing soil nutrient availability, which was of great significance for the successful introduction and cultivation of *D. sinicus* [11,58]. 

### 3.4. Potential Key Microbes Associated with Rapid Growth of D. sinicus 

Hundreds of woody bamboo species occur on the red soil and are characterized by their rapid growth [30]. Previous studies showed significant upregulation of genes associated with cell elongation or division, hormone signal transduction and cell wall development during the rapid growth period of woody bamboo [26,31,59]. However, there were few studies on the relationship between red soil micro-organisms and rapid growth of woody bamboos. Red soil was characterized by low nutrient availability [2], but *D. sinicus* growing on the red soil exhibited a substantial aboveground biomass [30]. It was therefore of interest to see how the barren red soil provided sufficient available nutrients for *D*. *sinicus* during its rapid growth period. This study revealed that the NH_4_^+^-N contents in the rhizosphere soil of all types were significantly higher in August than in May (*p* < 0.05) (Table 1), indicating substantial NH_4_^+^-N absorption by *D. sinicus* during the shooting and rapid growth period, which was consistent with other bamboo species [35]. The correlation analysis highlighted a positive association between soil NH_4_^+^-N content and bacterial genera *Reyranella*, *Paenibacillus*, *Micromonospora*, *Nocardioides*, *Luedemannella*, *Pedomicrobium*, *Bradyrhizobium*, as well as fungal genera *Trichoderma*, *Mortierella*, *Trechispora*, *Apiotrichum*, *Exophiala*, *Acremonium*, *Cordana* (*p* < 0.05). Among them, the taxa associated with N fixation, such as *Bradyrhizobium*, *Paenibacillus* and *Trichoderma*, may contribute to increased soil NH_4_^+^-N availability [12,14,60,61], which could facilitate the rapid growth of *D. sinicus*. 

## 4. Materials and Methods

### 4.1. Soil Microbes Associated with Rapid Growth of D. sinicus 

The study examined three types of *D. sinicus*: bending, straight and introduced straight type. The habitat conditions for these types are presented in Table 2. Monthly precipitation data for the three locations were obtained from the WorldClim website (https://www.worldclim.org/data/worldclim21.html, accessed on 27 February 2022) (Appendix A). The introduced straight type in Xinping County originated from the straight type in Ximeng County in 2010, and the seedlings were cultivated following the methods of asexual reproduction, i.e., stem cutting and divided clumps. A close-to-nature management approach was adopted for the cultivation of bamboo forest. The sample plot under investigation represented the largest surviving population subsequent to the introduction of the straight type.

### 4.2. Soil Sample Collection and Investigation of Bamboo Biological Characteristics 

Soil sampling was conducted in May and August 2022, corresponding to the dormancy and shooting/rapid growth periods of the underground rhizome, respectively. Five quadrats (40 × 40 m) were established for each type of *D*. *sinicus*. Five healthy bamboo clumps were selected within each quadrat for the study, and the DBH of adult bamboos was measured. In August, the shooting rate was assessed by enumerating the number of adult bamboos and young shoots within each cluster. For each sampling event, five healthy bamboo clumps were excavated. After the loosely bound soil was shaken off, the tightly adhered soils were collected and used as rhizosphere soil. The collected fresh soils from each quadrat were sieved through a 2 mm mesh size and homogenized to create mixed samples. These mixed soil samples were divided into three parts. The first part was stored in sterilized centrifuge tubes in dry ice during transportation, then kept in an ultra-low-temperature refrigerator (−80 °C) at the laboratory for subsequent microbial diversity sequencing. The second part was stored in a portable refrigerator and stored at −20 °C for soil chemical property analysis after being transported to the laboratory. The third part of the soil sample (ca. 20 g) was promptly packed into aluminum boxes, weighed and transported back to the laboratory for moisture content determination [11]. 

### 4.3. The Measurement of Soil Properties 

Soil pH was measured using a pH meter (FE20K, Mettler-Toledo, Switzerland) at a ratio of 2.5:1 (water:soil). SOC was quantified using an automatic analyzer (Shimadzu, Kyoto, Japan). Total phosphorus (TP) was determined via molybdenum-antimony colorimetry. AP was extracted using the hydrochloric acid–sulfuric acid method. Total potassium (TK) and AK were measured using the atomic absorption photometry method. SWC was determined by drying the samples at 105 °C until constant weight [62]. Total nitrogen (TN), NH_4_^+^-N and NO_3_^−^-N were measured using the SEAL AutoAnalyzer 3 (Seal Analytical, Hamburg, Germany). 

### 4.4. DNA Extraction, Amplification and Sequencing 

Total DNA was extracted from 0.5 g of homogenized fresh soil samples using the PowerSoil DNA Isolation Kit (Mo Bio Laboratories Inc., Carlsbad, CA, USA). The quality and concentration of the DNA were examined using a NanoDrop ND-2000 spectrophotometer (Thermo Scientific, Wilmington, DE, USA). The extracted DNA was stored at −20 °C prior to analysis. To identify the characteristics of soil bacterial communities, the V3-V4 variable region of the 16S rRNA gene was amplified. The following primers were used: 338F (5′-ACTCCTACGGGAGGCAGCAG-3′) and 806R (5′-GGACTACHVGGGTWTCTAAT-3′) [63]. The PCR reaction system had a volume of 20 μL, consisting of 4 μL of 5 × FastPfu buffer, 2 μL of 2.5 mM dNTPs, 0.8 μL of each primer (5 μM), 0.4 μL of FastPfu Polymerase, 0.2 μL of BSA and 10 ng of Template DNA, with the rest being ddH_2_O. Meanwhile, fungal communities were characterized by amplifying the internal transcribed spacer 1 (ITS1) region using the universal eukaryotic primers ITS1F (5′-CTTGGTCATTTAGAGGAAGTAA-3′) and ITS2R (5′-GCTGCGTTCTTCATCGATGC-3′) [62]. The PCR reaction system (20 μL) included 2 μL of 10 × Buffer, 2 μL of 2.5 mM dNTPs, 0.8 μL of each primer (5 μM), 0.2 μL of rTaq Polymerase, 0.2 μL of BSA and 10 ng of Template DNA. PCR amplification was performed using the GeneAmp PCR System 9700 (Applied Biosystems, Foster City, CA, USA) under the following thermal cycling conditions: 95 °C for 3 min, 35 cycles at 95 °C for 30 s, 55 °C for 30 s, 72 °C for 45 s and a final extension at 72 °C for 10 min. Sequencing was performed on an Illumina MiSeqPE300 Platform (Illumina, San Diego, CA, USA) by Majorbio Bio-pharm Technology Co., Ltd. (Shanghai, China). The raw sequencing data were deposited in the NCBI Sequence Reading Archive (SRA), with the BioProject number PRJNA980898 for bacterial diversity and PRJNA980913 for fungal diversity. 

### 4.5. Data Analyses 

The FLASH (v1.2.11) software was utilized for merging raw paired-end sequences [64], and the fastp software (v0.19.6) was utilized for quality control [65]. Operational taxonomic units (OTUs) were classified using the Uparse software (v7.0) with a 97% similarity criterion [66]. The taxonomic assignment of representative OTUs was performed by the RDP classifier [67] based on the Silva database (https://www.arb-silva.de/, accessed on 9 December 2022) and the Unite database (https://unite.ut.ee/, accessed on 9 December 2022). Soil microbial community α diversity was evaluated using the Sob, Shannon and Chao1 indices. To analyze the β diversity of the microbial community, principal coordinate analysis (PCoA) based on the abund-jaccard distance algorithm was conducted. Within-group and between-group similarity comparisons were assessed using the analysis of similarity (ANOSIM). 

The composition of the microbial community was analyzed at the phylum and genus levels using taxonomic information. The LEfSe tool was used to determine the significant differences among samples [68], and the LDA was set to 4. The trophic modes of the fungal community were discerned based on FUNGuild [69]. The bacterial ecological functions were predicted using FAPROTAX [70], and bacterial phenotypes were predicted using the BugBase tool. The predicted results among groups were analyzed using the Kruskal–Wallis rank test. Soil properties with low multicollinearity, determined through the variance inflation factor (VIF) analysis, were retained. To explore the relationship between the microbial community and retained soil properties, the redundancy analysis (RDA) was conducted. The correlation between microbial taxa at the genus level and soil properties was assessed using the Spearman correlation coefficient. The calculation method for the bamboo stem weight (BSW) was as follows: BSW = 0.4531 × DBH^1.7961^ [30]. Statistical significance among samples was calculated using Duncan’s multiple comparison test. All statistical analyses were performed using R version 4.2.3. 

## 5. Conclusions

Based on amplification sequencing and measurements of soil properties in different types of *D. sinicus* at the rapid growth and dormancy periods of rhizosphere soil, this study suggested that *D. sinicus* exhibited an absorption preference for NH_4_^+^-N, particularly during the shooting and following the rapid growth period. Moreover, compared with the native habitat of the straight type, the soil bacterial functions of the introduced straight type exhibited a significant decrease in nitrogen fixation, and the dominant microbial function was associated with ureolysis. Saprophytic fungi were the dominant fungal functional taxa in both straight and introduced straight type soils. This may reflect the fact that bacterial communities were more susceptible to soil conditions compared to fungal communities. This paper constitutes one of the initial research works exploring the effects of rhizosphere micro-organisms in woody bamboo on its adaptability and rapid growth. When introducing and cultivating woody bamboo, it is imperative to not only select superior seed sources but to also consider the influence of microbes on the bamboo forest in order to enhance adaptation and growth. 

## Figures and Tables

**Figure 1 ijms-24-14665-f001:**
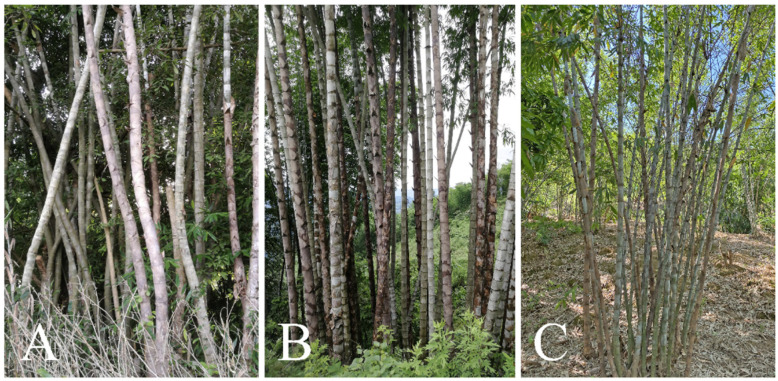
Morphological characteristics of bending type (**A**), straight type (**B**) and introduced straight type (**C**). Photographs by Peitong Dou.

**Figure 2 ijms-24-14665-f002:**
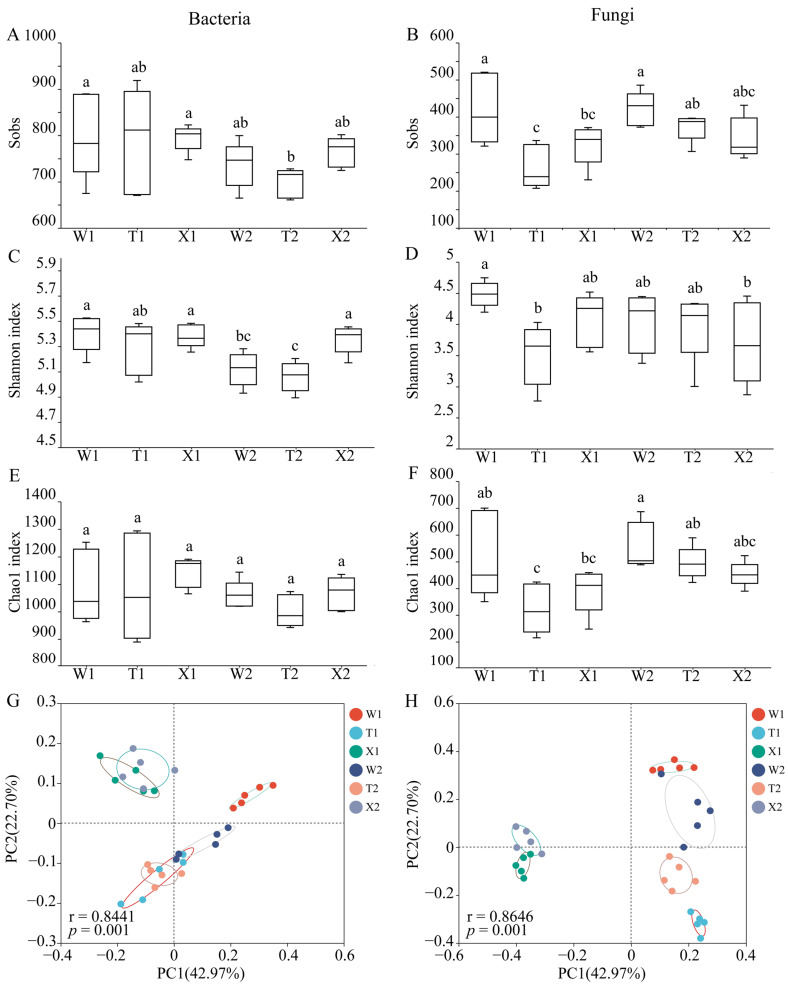
Rhizosphere microbial community α and β diversity in different soil samples. (**A**,**B**) Sobs of bacterial (**A**) and fungal (**B**) communities. (**C**,**D**) Shannon indexes of bacterial (**C**) and fungal (**D**) communities. (**E**,**F**) Chao1 indexes of bacterial (**E**) and fungal (**F**) communities. (**G**,**H**) Principal coordinate analysis (PCoA) of bacterial (**G**) and fungal (**H**) communities based on the abund-jaccard distance algorithm. W1, rhizosphere soil sample of bending type in May; W2, rhizosphere soil sample of bending type in August; T1, rhizosphere soil sample of straight type in May; T2, rhizosphere soil sample of straight type in August; X1, rhizosphere soil sample of introduced straight type in May; X2, rhizosphere soil sample of introduced straight type in August. Different letters in the same line indicate significant differences (*p* < 0.05).

**Figure 3 ijms-24-14665-f003:**
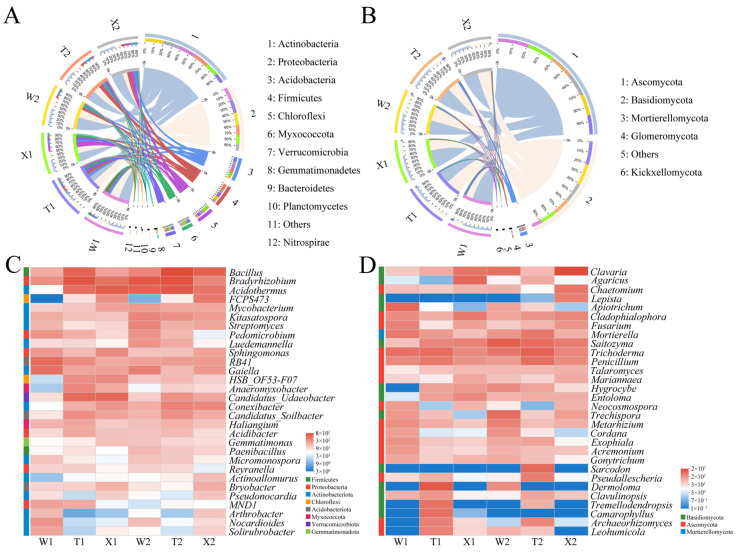
Microbial community composition in rhizosphere soil samples. Abundances of main bacterial (**A**) and fungal (**B**) community phyla. Heatmaps showing the top 30 abundant bacterial (**C**) and fungal (**D**) genera. W1, rhizosphere soil sample of bending type in May; W2, rhizosphere soil sample of bending type in August; T1, rhizosphere soil sample of straight type in May; T2, rhizosphere soil sample of straight type in August; X1, rhizosphere soil sample of introduced straight type in May; X2, rhizosphere soil sample of introduced straight type in August.

**Figure 4 ijms-24-14665-f004:**
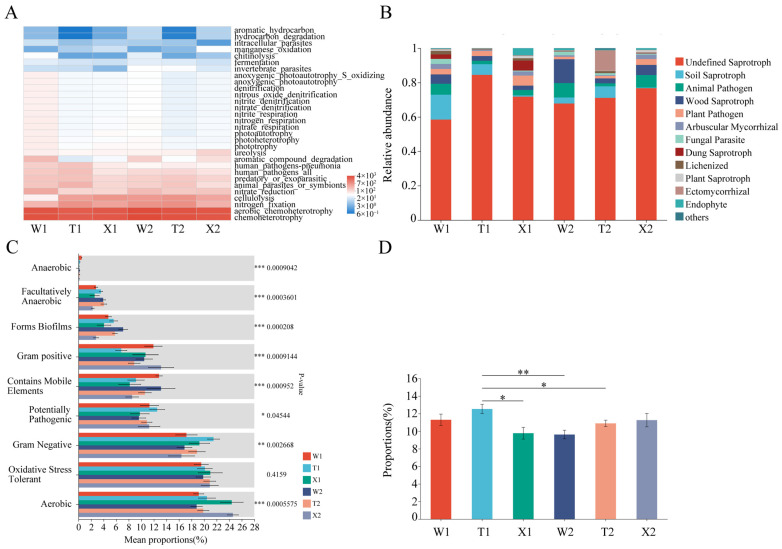
Functional prediction of bacteria (**A**) and fungi (**B**) in six rhizosphere soil samples. Significant difference test among groups in bacterial phenotypes (**C**). Kruskal−Wallis H−test for potentially pathogenic (**D**). W1, rhizosphere soil sample of bending type in May; W2, rhizosphere soil sample of bending type in August; T1, rhizosphere soil sample of straight type in May; T2, rhizosphere soil sample of straight type in August; X1, rhizosphere soil sample of introduced straight type in May; X2, rhizosphere soil sample of introduced straight type in August. * Significant differences at *p* < 0.05 level, ** Significant differences at *p* < 0.01 level, *** Significant differences at *p* < 0.001 level.

**Figure 5 ijms-24-14665-f005:**
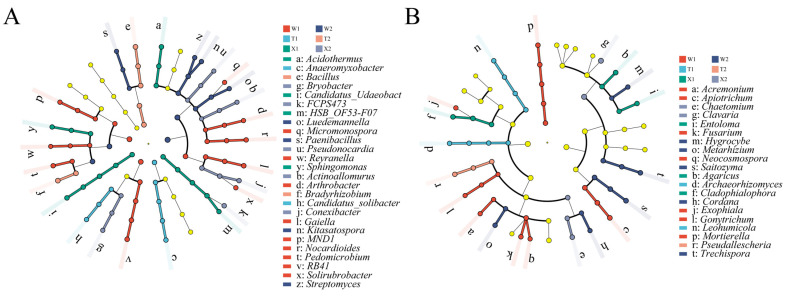
LEfSe analysis of microbial abundance in six rhizosphere soil samples. Cladogram showing taxa with different abundance values of bacterial (**A**) and fungal (**B**) community. W1, rhizosphere soil sample of bending type in May; W2, rhizosphere soil sample of bending type in August; T1, rhizosphere soil sample of straight type in May; T2, rhizosphere soil sample of straight type in August; X1, rhizosphere soil sample of introduced straight type in May; X2, rhizosphere soil sample of introduced straight type in August.

**Figure 6 ijms-24-14665-f006:**
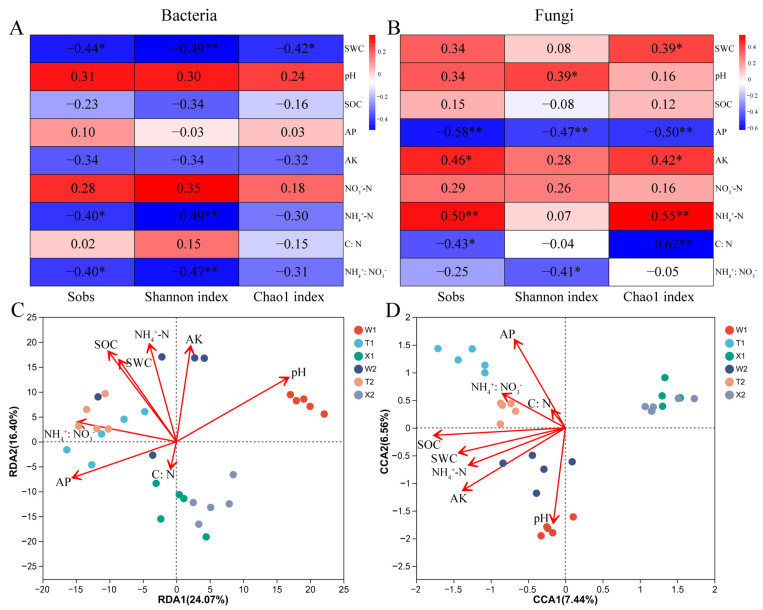
Correlations of selected soil properties with bacterial (**A**) or fungal (**B**) α diversity indices. Redundancy analysis of soil properties and bacterial (**C**) or fungal (**D**) community in rhizosphere of *D*. *sinicus*. Color depth represented the magnitude of correlation R value. * 0.01  < *p* ≤ 0.05; ** 0.001 < *p* ≤ 0.01. SWC, soil water content; SOC, soil organic carbon; AP, available phosphorus; AK, available potassium; NO_3_^−^-N, nitrate nitrogen; NH_4_^+^-N, ammonia nitrogen; C: N, the ratio between soil organic carbon and soil total nitrogen; NH_4_^+^:NO_3_^−^, the ratio between ammonia nitrogen and nitrate nitrogen; W1, rhizosphere soil sample of bending type in May; W2, rhizosphere soil sample of bending type in August; T1, rhizosphere soil sample of straight type in May; T2, rhizosphere soil sample of straight type in August; X1, rhizosphere soil sample of introduced straight type in May; X2, rhizosphere soil sample of introduced straight type in August.

**Figure 7 ijms-24-14665-f007:**
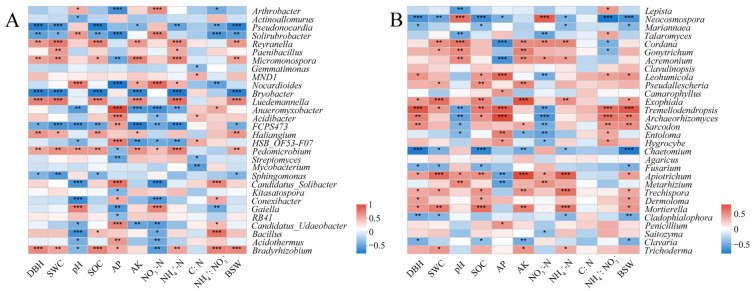
Heatmaps showing correlations between soil factors, biological characteristics and the first 30 genera of bacteria (**A**) and fungi (**B**). Color depth represented the magnitude of correlation R value. * 0.01  < *p* ≤ 0.05; ** 0.001 < *p* ≤ 0.01; *** *p* ≤ 0.001. SWC, soil water content; SOC, soil organic carbon; AP, available phosphorus; AK, available potassium; NO_3_^−^-N, nitrate nitrogen; NH_4_^+^-N, ammonia nitrogen; C: N, the ratio between soil organic carbon and soil total nitrogen; NH_4_^+^: NO_3_^−^, the ratio between ammonia nitrogen and nitrate nitrogen; DBH, diameter at breast height; BSW, bamboo stem weight.

**Figure 8 ijms-24-14665-f008:**
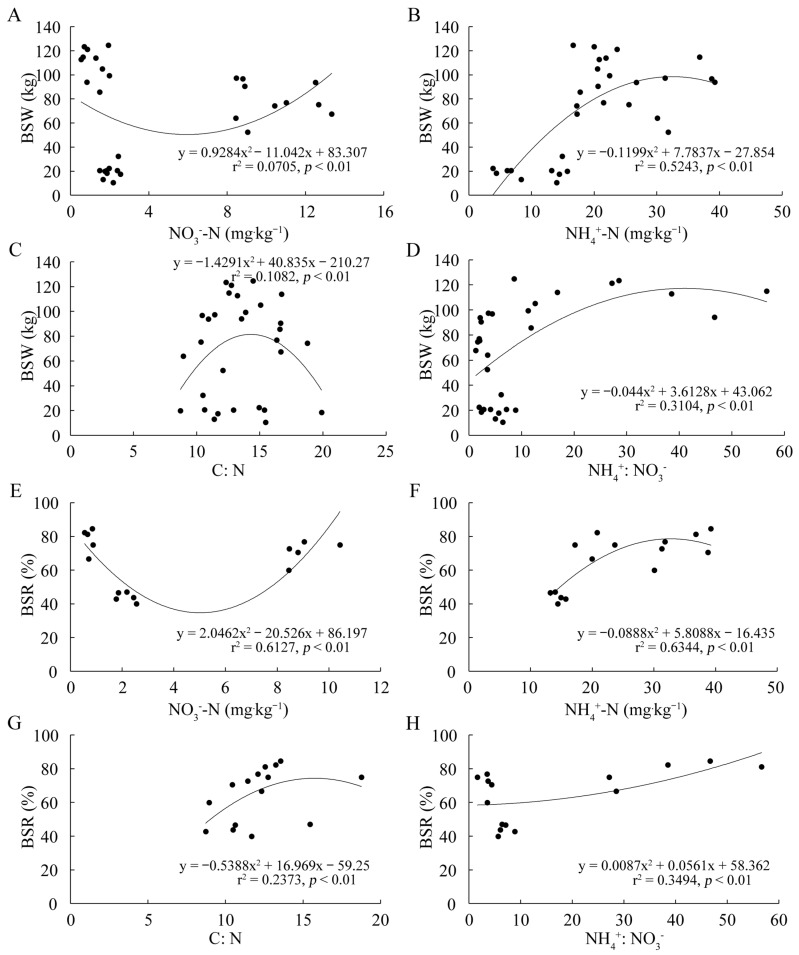
The regression relation between selected soil properties and the biological characteristics of *D. sinicus*. Regression analysis between BSW and soil NO_3_^−^-N content (**A**), NH_4_^+^-N content (**B**), C: N (**C**), and NH_4_^+^: NO_3_^−^ (**D**). Regression analysis between BSR and soil NO_3_^−^-N content (**E**), NH_4_^+^-N content (**F**), C: N (**G**), and NH_4_^+^: NO_3_^−^ (**H**). NO_3_^−^-N, nitrate nitrogen; NH_4_^+^-N, ammonia nitrogen; C: N, the ratio between soil organic carbon and soil total nitrogen; NH_4_^+^: NO_3_^−^, the ratio between ammonia nitrogen and nitrate nitrogen; BSW, bamboo stem weight; BSR, bamboo shooting ratio.

**Table 1 ijms-24-14665-t001:** Soil properties and biological characteristics of each *D. sinicus* type in different periods.

	W1	T1	X1	W2	T2	X2
pH	6.13 ± 0.0231 a	5.66 ± 0.0182 d	5.74 ± 0.0130 c	6.02 ± 0.0114 b	5.58 ± 0.0180 e	5.62 ± 0.0100 de
SOC (g·kg^−1^)	24.88 ± 1.7468 c	30.40 ± 1.4859 ab	15.30 ± 1.3172 d	26.64 ± 1.6675 bc	32.06 ± 1.5740 a	10.31 ± 0.7668 e
AP (mg·kg^−1^)	4.38 ± 0.4409 d	13.94 ± 1.0245 a	9.12 ± 0.3652 b	4.82 ± 0.5472 d	8.68 ± 0.6967 b	6.76 ± 0.4336 c
TP (g·kg^−1^)	1.29 ± 0.1118 a	1.28 ± 0.2034 a	0.44 ± 0.0089 b	1.28 ± 0.1660 a	1.19 ± 0.1282 a	0.59 ± 0.0551 b
AK (mg·kg^−1^)	297.0 ± 6.9498 a	167.6 ± 7.8333 c	58.8 ± 3.2465 d	225.2 ± 16.7404 b	289.2 ± 26.0776 a	78.0 ± 7.2732 d
TK (g·kg^−1^)	5.60 ± 0.2074 c	11.78 ± 0.2177 a	7.34 ± 0.1568 b	5.30 ± 0.3795 c	11.22 ± 0.2311 a	7.00 ± 0.3521 b
TN (g·kg^−1^)	1.85 ± 0.2517 b	1.99 ± 0.1316 ab	1.04 ± 0.0592 c	2.34 ± 0.3391 ab	2.49 ± 0.1058 a	0.92 ± 0.0496 c
NO_3_^−^-N (mg·kg^−1^)	11.69 ± 0.7936 a	1.67 ± 0.1309 cd	1.88 ± 0.1541 c	9.04 ± 0.3649 b	0.72 ± 0.0611 d	2.16 ± 0.1548 c
NH_4_^+^-N (mg·kg^−1^)	22.32 ± 1.7144 bc	19.85 ± 1.1401 cd	5.89 ± 0.8136 e	29.83 ± 3.5011 a	28.10 ± 4.1165 ab	14.46 ± 0.4222 d
SWC (%)	25.79 ± 0.9174 b	26.29 ± 1.7114 b	15.55 ± 0.5985 c	30.30 ± 2.0609 ab	33.17 ± 2.5209 a	20.01 ± 0.6092 c
C: N	14.19 ± 1.4521 ab	15.34 ± 0.5688 a	14.90 ± 1.4384 ab	12.33 ± 1.6959 ab	12.88 ± 0.2239 ab	11.39 ± 1.1213 b
NH_4_^+^: NO_3_^−^	1.94 ± 0.1736 c	12.22 ± 1.3341 b	3.24 ± 0.5697 c	3.37 ± 0.4578 c	39.52 ± 5.5597 a	6.85 ± 0.5630 bc
DBH (cm)	17.90 ± 0.6058 b	20.79 ± 0.7266 a	7.99 ± 0.3923 c	17.34 ± 1.1352 b	21.61 ± 0.5640 a	8.18 ± 0.8046 c
BSW (kg)	80.90 ± 4.9101 b	105.81 ± 6.5896 a	19.06 ± 1.6011 c	77.09 ± 8.9063 b	113.33 ± 5.1874 a	20.30 ± 3.5416 c
BSR (%)	-	-	-	71.05 ± 2.9599 a	77.98 ± 3.2459 a	44.07 ± 1.3003 b

Values are means ± SE; Different letters in the same line indicate significant differences (*p* < 0.05); W1, rhizosphere soil sample of bending type in May; W2, rhizosphere soil sample of bending type in August; T1, rhizosphere soil sample of straight type in May; T2, rhizosphere soil sample of straight type in August; X1, rhizosphere soil sample of introduced straight type in May; X2, rhizosphere soil sample of introduced straight type in August. SOC, soil organic carbon; AP, available phosphorus; TP, soil total phosphorus; AK, available potassium; TK, soil total potassium; TN, soil total nitrogen; NO_3_^−^-N, nitrate nitrogen; NH_4_^+^-N, ammonia nitrogen; SWC, soil water content; C: N, the ratio between soil organic carbon and soil total nitrogen; NH_4_^+^: NO_3_^−^, the ratio between ammonia nitrogen and nitrate nitrogen; DBH, diameter at breast height. BSW, bamboo stem weight; BSR, bamboo shooting ratio.

**Table 2 ijms-24-14665-t002:** The habitat conditions of different types of *D. sinicus*.

The Type of *D*. *sinicus*	Study Site	Altitude/m	Climate Type	Annual Average Temperature/°C	Annual Precipitation/mm	Frost-Free Period/Day	Soil Type
Bending type	Menglian County, YN (22.20 N, 99.33 E)	1040	south subtropical climate	19.7	1363.6	300	lateritic red soil
Straight type	Ximeng County, YN (22.63 N, 99.62 E)	1122	subtropical mountain humid monsoon climate	19.1	1629.2	362	lateritic red soil
Introduced straight type	Yuxi County, YN (24.12 N, 102.63 E)	878	subtropical semi-humid plateau monsoon climate	19.0	950.0	315	red soil

## Data Availability

The data supporting the findings of this study are available on request from the corresponding author.

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
