# Peer review of "Rhizosphere Microbe Affects Soil Available Nitrogen and Its Implication for the Ecological Adaptability and Rapid Growth of Dendrocalamus sinicus, the Strongest Bamboo in the World"

_ijms, 2023, doi:10.3390/ijms241914665_

Round 1
Reviewer 1 Report
The research presented in the manuscript on the study of the influence of the soil microbial community on the development and success of the cultivation of woody bamboos and the determination of the characteristics of nitrogen absorption and microorganisms that favor the rapid growth of bamboo are very interesting. They have a high cognitive value, but also practical, they can be used when setting up bamboo plantations and choosing the right place and method of cultivation. The experiments presented in the manuscript were properly planned, it allows to explain the hypotheses put forward in the introduction. The collected measurements and analysis results are very extensive. The results were statistically analyzed and discussed and interpreted in detail. Tables and graphs are generally well prepared, understandable and legible.
I have only one comment on the manuscript.
I couldn't find tables and figures in the supplementary files. There is only 'Change of authorship forms'
Author Response
Cover letter
Dear reviewers:
Thank you very much for taking the time to review this manuscript. Please find the detailed responses below and the corresponding revisions/corrections highlighted/in track changes in the re-submitted files. Your concerns were in careful consideration and response in best of detail. The revisions to the manuscript had been marked in red.
Best regards,
Hanqi Yang
Questions for General Evaluation |
Reviewer’s Evaluation |
Response and Revisions |
Does the introduction provide sufficient background and include all relevant references? |
Yes |
|
Are all the cited references relevant to the research? |
Yes |
|
Is the research design appropriate? |
Yes |
|
Are the methods adequately described? |
Yes |
|
Are the results clearly presented? |
Can be improved |
The results and conclusions of the manuscript have been revised. |
Are the conclusions supported by the results? |
Yes |
|
Comments 1: The research presented in the manuscript on the study of the influence of the soil microbial community on the development and success of the cultivation of woody bamboos and the determination of the characteristics of nitrogen absorption and microorganisms that favor the rapid growth of bamboo are very interesting. They have a high cognitive value, but also practical, they can be used when setting up bamboo plantations and choosing the right place and method of cultivation. The experiments presented in the manuscript were properly planned, it allows to explain the hypotheses put forward in the introduction. The collected measurements and analysis results are very extensive. The results were statistically analyzed and discussed and interpreted in detail. Tables and graphs are generally well prepared, understandable and legible.
I have only one comment on the manuscript.
I couldn't find tables and figures in the supplementary files. There is only 'Change of authorship forms'.
Response 1: Thank you for your recognition of this study. The tables and figures in the manuscript have been re-uploaded.
In addition, we have made modifications to the following content:
In citations of the manuscript, the citation ‘[19,20,21,22]’ was revised to ‘[19-22]’ (line 52).
The citation ‘[32,26,35]’ was revised to ‘[26,32,35]’ (line 87).
The citation ‘[29,31]’ was revised to ‘[30,31]’ (line 379).
The citation ‘[51,13,40]’ was revised to ‘[10,38,51]’ (line 397).
The citation ‘[32]’ was revised to ‘[29]’ (line 399).
The citation ‘[40]’ was revised to ‘[38]’ (line 401).
The citation ‘[14]’ was revised to ‘[11]’ (line 404).
The citation ‘[22,23]’ was revised to ‘[19,20]’ (line 438).
The citation ‘[22]’ was revised to ‘[19]’ (line 441).
The citation ‘[14,61]’ was revised to ‘[11,61]’ (line 449).
The citation ‘[29,30]’ was revised to ‘[30,31]’ (line 452).
The citation ‘[35,37,62]’ was revised to ‘[26,32,35,62]’ (line 454).
The citation ‘[40]’ was revised to ‘[38]’ (line 462).
In the abstract, the partial content ‘The result showed that each type of D. sinicus preferred to absorb ammonia nitrogen (NH4+-N) rather than nitrate nitrogen (NO3--N), and required more NH4+-N at germination or rapid growth period than the dormancy period. Microbial taxa increasing NH4+-N availability facilitated the rapid growth of young shoots. Totally, nitrogen fixation capacity of soil bacteria in the straight type was significantly higher than that in the introduced straight type, while the ureolysis capacity had an opposite trend’ was revised to ‘The result showed that each type of D. sinicus preferred to absorb ammonia nitrogen (NH4+-N) rather than nitrate nitrogen (NO3--N), and required more NH4+-N at germination or rapid growth period than the dormancy period. Totally, nitrogen fixation capacity of soil bacteria in the straight type was significantly higher than that in the introduced straight type, while the ureolysis capacity had an opposite trend’ (lines 20 to 22); The word ‘promote’ was revised to ‘affect’ (line 24).
In the introduction, the partial content ‘largest’ was revised to ‘strongest’ (line 77).
The ‘soil microbial communities’ was revised to ‘micro-ecological factors’ (line 94).
The ‘microbial taxa’ was revised to ‘potential microbial taxa’ (line 96).
In the results, the partial content ‘across all stages’ was revised to ‘at two periods’ (line 117).
The format of Table 1 was adjusted (line 132).
The font format of lines 133 to 143 was revised (lines 133 to 143).
The Figure 3 was revised and uploaded (line 224).
The sentence 'Consequently, the soil NH4+-N content played a significant role in shaping the biological traits of D. sinicus, such as BSW and BSR’ was revised to ‘This result indicated that the NH4+-N availability had a greater impact on the formation of biological traits of D. sinicus’ (line 330, 331).
In the discussion, the partial content ‘This suggested that a higher NH4+: NO3- was more crucial for the rapid growth of D. sinicus compared to the dormancy period of the underground rhizome’ was revised to ‘This result suggested that a higher NH4+: NO3- was more crucial for the rapid growth of D. sinicus compared to the dormancy period of the underground rhizome’ (line 359).
The ‘carbon to nitrogen ratio’ was ‘C: N’ (line 369).
The partial content ‘Rhizosphere microbes improve NH4+-N absorption of D. sinicus in red soil’ was revised to ‘Influence of Rhizosphere microbes on NH4+-N availability in red soil’ (line 377).
The partial content ‘In this study, a significant positive correlation was observed between soil NH4+-N content and the bacterial genera Bradyrhizobium and Paenibacillus, as well as the fungal genus Trichoderma (p < 0.05) (Figure 7). Meanwhile, functional predictions of the bacterial community showed dominant roles of nitrogen fixation, cellulolysis, and ureolysis in the rhizosphere soil of D. sinicus, indicating a high capacity for carbohydrate conversion and biological nitrogen fixation [14]. Meanwhile, FUNGuild analysis revealed a high proportion of saprotrophic fungi, which facilitated to increase the availability of carbon and nitrogen [52,53]’ was revised to ‘In this study, the prediction results of bacterial functions showed dominant roles of nitrogen fixation, cellulolysis, and ureolysis in the rhizosphere soil of D. sinicus, indicating a high capacity for carbohydrate conversion and biological nitrogen fixation [11]. Remarkably, the rhizosphere microbes of straight type possessed higher nitrogen fixing capacity, which could lead to higher soil NH4+-N content compared to the introduced straight type (Table 1, Table S3) [51]. Furthermore, FUNGuild analysis revealed a high proportion of saprotrophic fungi, which facilitated to increase the availability of carbon and nitrogen [52,53]’ (lines 401 to 409).
The partial content ‘Soil microbes contribute to the fitness of D. sinicus’ was revised to ‘Influence of micro-ecological factors on the adaptation of D. sinicus’ (line 410).
The sentence ‘The rhizosphere microbial community can affect the fitness of host plants [54]’ was revised to ‘The rhizosphere micro-ecology of plants is influenced by the secretion of secondary metabolites, the taxa of rhizosphere microbes, and the physicochemical properties of the soil. The disruption of micro-ecological balance can result in soil degradation, thereby influence plant growth [54]’ (lines 411 to 414).
The sentence ‘Previous studies indicated that plant adaptability was influenced by climate and soil conditions, and similar climate and soil characteristics had a minimal impact on the abiotic factors between the original and the introduced sites [55,56]’ was revised to ‘Previous studies indicated that plant adaptability was influenced by soil conditions and rhizosphere microbial taxa [55,56]’ (line 414, 415).
The partial content ‘In this study, apart from a significantly lower Shannon index of bacterial community observed in August for the introduced straight type compared to the straight type, no statistically significant differences were found in other indexes between the two types (Figure 2). These findings suggested that a similarity in soil microbial communities played a crucial role in the successful introduction of the straight type. Although the introduced straight type had the same origin region as the straight type, the introduced straight type exhibited significantly smaller DBH, BSW and BSR (p < 0.05) (Table 1)’ was revised to ‘(In this study, although the introduced straight type had the same origin region as the straight type, the introduced straight type exhibited significantly smaller DBH, BSW and BSR (p < 0.05) (Table 1))’ (lines 415 to 418).
The ‘indicated’ was revised to ‘indirectly suggested’ (line 420).
The partial content ‘Meanwhile, the prediction results of soil bacterial functions showed that nitrogen fixation abundance was significantly lower in the introduced straight type compared to the straight type at each stage (Figure 4A, Table S3), which may be an important factor leading to the low soil NH4+-N content of the introduced straight type [51]’ was revised to ‘Meanwhile, apart from a significantly higher Shannon index of bacterial community observed in August for the introduced straight type compared to the straight type, no statistically significant differences were found in other indexes between the two types (Figure 2). The prediction results of soil bacterial functions showed that nitrogen fixation abundance was significantly lower in the introduced straight type compared to the straight type at each stage (Figure 4A, Table S3), which may be an important factor leading to the low soil NH4+-N content of the introduced straight type [51]’ (lines 422 to 429).
The partial content ‘The plant-microbe interactions were likely to influence fitness of plant, as supported by previous research’ was revised to ‘The interaction among soil, plants and soil microbes was likely to influence adaptation of plant, as supported by previous research’ (line 436, 437).
The partial content ‘Soil microbes associated with rapid growth of D. sinicus’ was revised to ‘Potential key microbes associated with rapid growth of D. sinicus’ (line 450).
The partial content ‘Correlation analysis highlighted a positive association between soil NH4+-N content and bacterial genera Reyranella, Paenibacillus, Micromonospora, Nocardioides, Luedemannella, Pedomicrobium, Bradyrhizobium (p < 0.05), as well as fungal genera Trichoderma, Mortierella, Trechispora, Apiotrichum, Exophiala, Acremonium, Cordana. Among them, Bradyrhizobium and Paenibacillus from bacterial communities, and Trichoderma from fungal communities may contribute to increased NH4+-N availability for D. sinicus [15,17,64,65]. The enrichment of these microorganisms was a direct response to the increased demand for NH4+-N during rapid growth of D. sinicus’ was revised to ‘Correlation analysis highlighted a positive association between soil NH4+-N content and bacterial genera Reyranella, Paenibacillus, Micromonospora, Nocardioides, Luedemannella, Pedomicrobium, Bradyrhizobium, as well as fungal genera Trichoderma, Mortierella, Trechispora, Apiotrichum, Exophiala, Acremonium, Cordana (p < 0.05). Among them, the taxa associated with N fixation, such as Bradyrhizobium, Paenibacillus, and Trichoderma, may contribute to increased soil NH4+-N availability [12,14,64,65], which could facilitate the rapid growth of D. sinicus’ (lines 463 to 469).
In the materials and methods, the format of Table 2 was adjusted (line 481).
In the conclusions, the word ‘suggests’ was revised to ‘suggested’; The word ‘exhibits’ was revised to ‘exhibited’ (line 560).
The partial content ‘Bacterial genera Bradyrhizobium and Paenibacillus, as well as the fungal genus Trichoderma, may contribute to the improvement of NH4+-N content in the rhizosphere soil during the rapid growth period. Moreover, compared with the native habitat of the straight type, the soil microbial functions of the introduced straight type exhibited a significant decrease in nitrogen fixation, and dominant microbial function was associated with ureolysis. It may reflect that the soil properties have a great impact on rhizosphere microbial communities. Saprophytic fungi were the dominant fungal functional taxa in both straight and introduced straight type soils. Thus, the enrichment and functions of soil microbes play a crucial role in facilitating the rapid growth and adaptability of D. sinicus. when introducing and cultivating woody bamboo, it is imperative to not only select superior seed sources, but also consider the influence of microbes on the bamboo forest in order to enhance adaptation and growth’ was revised to ‘Moreover, compared with the native habitat of the straight type, the soil bacterial functions of the introduced straight type exhibited a significant decrease in nitrogen fixation, and dominant microbial function was associated with ureolysis. Saprophytic fungi were the dominant fungal functional taxa in both straight and introduced straight type soils. It may reflect that bacterial communities were more susceptible to soil conditions compared to fungal communities. This paper constitutes one of the initial researches exploring the effects of rhizosphere microorganisms in woody bamboo on its adaptability and rapid growth. When introducing and cultivating woody bamboo, it is imperative to not only select superior seed sources, but also consider the influence of microbes on the bamboo forest in order to enhance adaptation and growth’ (lines 561 to 571).
In the supplementary materials, the partial content ‘Abund_Jaccard’ was revised to ‘abund-jaccard’ (line 573).
In the references, the reference for [54] was revised from ‘Thiergart, T.; Durán, P.; Ellis, T.; Vannier, N.; Garrido-Oter, R.; Kemen, E.; Roux, F.; Alonso-Blanco, C.; Ågren, J.; Schulze-Lefert, P.; et al. Root microbiota assembly and adaptive differentiation among European Arabidopsis populations. Nat. Ecol. Evol. 2020, 4, 122-131. https://doi.org/10.1038/s41559-019-1063-3.’ to ‘Fang, X.Y.; Yu, T.B.; Yang, L.; Zang, H.D.; Zeng, Z.H; Yang, Y.D. Progress of bacterial wilt and its soil micro-ecological regulation in peanut. Chin. J. Eco-Agric. 2023. https://link.cnki.net/urlid/13.1432.S.20230815.1123.001.’ (line 729, 730).
The reference for [56] was revised from ‘Andrade, L.R.M.; Aquino, F.G.; Echevarria, G.; Oliveira, J.S.; Pereira, C.D.; Malaquias, J.V.; Souza, K.S.; Montargès-Pelletier, E.; Faleiro, F.G.; Junior, F.B.R.; et al. Edaphic factors as genetic selection agents and adaptation drivers of native plant species in harsh environments of the Brazilian savanna. Plant Soil. 2022, 479, 301-323. https://doi.org/10.1007/s11104-022-05520-3.’ to ‘Thiergart, T.; Durán, P.; Ellis, T.; Vannier, N.; Garrido-Oter, R.; Kemen, E.; Roux, F.; Alonso-Blanco, C.; Ågren, J.; Schulze-Lefert, P.; et al. Root microbiota assembly and adaptive differentiation among European Arabidopsis populations. Nat. Ecol. Evol. 2020, 4, 122-131. https://doi.org/10.1038/s41559-019-1063-3.’ (lines 734 to 736).

Reviewer 2 Report
In the work submitted for review, the authors sequenced soil microorganisms and studied the soil properties of three different types of Dendrocalamus sinicus.
Despite the large number of analyzes of the soil composition and the sequenced group of microorganisms, it is not very clear on what basis the conclusions are made about the beneficial effect of these organisms on bamboo, since significant biological characteristics, such as the protein content in the plant, which proves the amount of available N, have not been studied. The number of shoots and fresh weight is a bit far-fetched inference. There is no demonstrated direct effect of microbes on the rapid growth of plants, as shown in the title.
In addition, the text contains editorial, punctuation and substantive errors, e.g. Actinobacteria should be used,
Line 345 Plants with a lowercase letter,
line 373 NH+ and 4 low index etc.....
Author Response
Cover letter
Dear reviewers:
Thank you very much for taking the time to review this manuscript. Please find the detailed responses below and the corresponding revisions/corrections highlighted/in track changes in the re-submitted files. Your concerns were in careful consideration and response in best of detail. The revisions to the manuscript had been marked in red.
Best regards,
Hanqi Yang
Questions for General Evaluation |
Reviewer’s Evaluation |
Response and Revisions |
Does the introduction provide sufficient background and include all relevant references? |
Yes |
|
Are all the cited references relevant to the research? |
Yes |
|
Is the research design appropriate? |
Must be improved |
We have revised the abstract, introduction, results, and discussion to align with the current experimental design. |
Are the methods adequately described? |
Yes |
|
Are the results clearly presented? |
Yes |
|
Are the conclusions supported by the results? |
Must be improved |
The results and conclusions of the manuscript have been revised. |
Comments 1: In the work submitted for review, the authors sequenced soil microorganisms and studied the soil properties of three different types of Dendrocalamus sinicus.
Despite the large number of analyzes of the soil composition and the sequenced group of microorganisms, it is not very clear on what basis the conclusions are made about the beneficial effect of these organisms on bamboo, since significant biological characteristics, such as the protein content in the plant, which proves the amount of available N, have not been studied. The number of shoots and fresh weight is a bit far-fetched inference. There is no demonstrated direct effect of microbes on the rapid growth of plants, as shown in the title.
Response 1: Thank you for pointing this out. In response to your concerns, our plan is to consider stem weight and shoot rate as indirect measures to reflect this relationship between soil nitrogen content and rapid growth of woody bamboo through revising the abstract (line 18 to 24), results (line 330, 331), discussions (line 337, 401 to 409, 410 to 429, 436 to 438, 450, 463 to 469), and conclusions (line 561 to 571) of the manuscript. D. sinicus, is the strongest bamboo that has been recorded in the world, but its distribution area is narrow. There are currently no successful cases of large-scale introduction. This article is an initial and valuable study exploring the effects of soil microorganisms on the growth and adaptability of D. sinicus. Your suggestion provides us with a new perspective on our future research direction. Next, we will continue to search for more direct evidence of microorganisms affecting the rapid growth of woody bamboo.
Comments 2: In addition, the text contains editorial, punctuation and substantive errors, e.g. Actinobacteria should be used.
Response 2: The ‘Actinobacteriota’ in the manuscript was revised to ‘Actinobacteria’ (lines 176, 178, 182). In addition, some words in Figure 3 and Table S1 have also been revised. The ‘Proteobacteria’ was revised to ‘Proteobacteria’. The ‘Acidobacteriota’ was revised to ‘Acidobacteria’. The ‘Verrucomicrobiota’ was revised to ‘Verrucomicrobia’. The ‘Gemmatimonadota’ was revised to ‘Gemmatimonadetes’. The ‘Bacteroidota’ was revised to ‘Bacteroidetes’. The ‘Planctomycetota’ was revised to ‘Planctomycetes’. The ‘Nitrospirota’ was revised to ‘Nitrospirae’.
Comments 3: Line 345 Plants with a lowercase letter.
Response 3: The word ‘plant’ was revised to lowercase (line 349).
Comments 4: line 373 NH+ and 4 low index etc.....
Response 4: The writing of NH4+-N was revised (lines 339, 377).
In addition, we have made modifications to the following content:
In citations of the manuscript, the citation ‘[19,20,21,22]’ was revised to ‘[19-22]’ (line 52).
The citation ‘[32,26,35]’ was revised to ‘[26,32,35]’ (line 87).
The citation ‘[29,31]’ was revised to ‘[30,31]’ (line 379).
The citation ‘[51,13,40]’ was revised to ‘[10,38,51]’ (line 397).
The citation ‘[32]’ was revised to ‘[29]’ (line 399).
The citation ‘[40]’ was revised to ‘[38]’ (line 401).
The citation ‘[14]’ was revised to ‘[11]’ (line 404).
The citation ‘[22,23]’ was revised to ‘[19,20]’ (line 438).
The citation ‘[22]’ was revised to ‘[19]’ (line 441).
The citation ‘[14,61]’ was revised to ‘[11,61]’ (line 449).
The citation ‘[29,30]’ was revised to ‘[30,31]’ (line 452).
The citation ‘[35,37,62]’ was revised to ‘[26,32,35,62]’ (line 454).
The citation ‘[40]’ was revised to ‘[38]’ (line 462).
In the abstract, the partial content ‘The result showed that each type of D. sinicus preferred to absorb ammonia nitrogen (NH4+-N) rather than nitrate nitrogen (NO3--N), and required more NH4+-N at germination or rapid growth period than the dormancy period. Microbial taxa increasing NH4+-N availability facilitated the rapid growth of young shoots. Totally, nitrogen fixation capacity of soil bacteria in the straight type was significantly higher than that in the introduced straight type, while the ureolysis capacity had an opposite trend’ was revised to ‘The result showed that each type of D. sinicus preferred to absorb ammonia nitrogen (NH4+-N) rather than nitrate nitrogen (NO3--N), and required more NH4+-N at germination or rapid growth period than the dormancy period. Totally, nitrogen fixation capacity of soil bacteria in the straight type was significantly higher than that in the introduced straight type, while the ureolysis capacity had an opposite trend’ (lines 20 to 22); The word ‘promote’ was revised to ‘affect’ (line 24).
In the introduction, the partial content ‘largest’ was revised to ‘strongest’ (line 77).
The ‘soil microbial communities’ was revised to ‘micro-ecological factors’ (line 94).
The ‘microbial taxa’ was revised to ‘potential microbial taxa’ (line 96).
In the results, the partial content ‘across all stages’ was revised to ‘at two periods’ (line 117).
The format of Table 1 was adjusted (line 132).
The font format of lines 133 to 143 was revised (lines 133 to 143).
The Figure 3 was revised and uploaded (line 224).
The sentence 'Consequently, the soil NH4+-N content played a significant role in shaping the biological traits of D. sinicus, such as BSW and BSR’ was revised to ‘This result indicated that the NH4+-N availability had a greater impact on the formation of biological traits of D. sinicus’ (line 330, 331).
In the discussion, the partial content ‘This suggested that a higher NH4+: NO3- was more crucial for the rapid growth of D. sinicus compared to the dormancy period of the underground rhizome’ was revised to ‘This result suggested that a higher NH4+: NO3- was more crucial for the rapid growth of D. sinicus compared to the dormancy period of the underground rhizome’ (line 359).
The ‘carbon to nitrogen ratio’ was ‘C: N’ (line 369).
The partial content ‘Rhizosphere microbes improve NH4+-N absorption of D. sinicus in red soil’ was revised to ‘Influence of Rhizosphere microbes on NH4+-N availability in red soil’ (line 377).
The partial content ‘In this study, a significant positive correlation was observed between soil NH4+-N content and the bacterial genera Bradyrhizobium and Paenibacillus, as well as the fungal genus Trichoderma (p < 0.05) (Figure 7). Meanwhile, functional predictions of the bacterial community showed dominant roles of nitrogen fixation, cellulolysis, and ureolysis in the rhizosphere soil of D. sinicus, indicating a high capacity for carbohydrate conversion and biological nitrogen fixation [14]. Meanwhile, FUNGuild analysis revealed a high proportion of saprotrophic fungi, which facilitated to increase the availability of carbon and nitrogen [52,53]’ was revised to ‘In this study, the prediction results of bacterial functions showed dominant roles of nitrogen fixation, cellulolysis, and ureolysis in the rhizosphere soil of D. sinicus, indicating a high capacity for carbohydrate conversion and biological nitrogen fixation [11]. Remarkably, the rhizosphere microbes of straight type possessed higher nitrogen fixing capacity, which could lead to higher soil NH4+-N content compared to the introduced straight type (Table 1, Table S3) [51]. Furthermore, FUNGuild analysis revealed a high proportion of saprotrophic fungi, which facilitated to increase the availability of carbon and nitrogen [52,53]’ (lines 401 to 409).
The partial content ‘Soil microbes contribute to the fitness of D. sinicus’ was revised to ‘Influence of micro-ecological factors on the adaptation of D. sinicus’ (line 410).
The sentence ‘The rhizosphere microbial community can affect the fitness of host plants [54]’ was revised to ‘The rhizosphere micro-ecology of plants is influenced by the secretion of secondary metabolites, the taxa of rhizosphere microbes, and the physicochemical properties of the soil. The disruption of micro-ecological balance can result in soil degradation, thereby influence plant growth [54]’ (lines 411 to 414).
The sentence ‘Previous studies indicated that plant adaptability was influenced by climate and soil conditions, and similar climate and soil characteristics had a minimal impact on the abiotic factors between the original and the introduced sites [55,56]’ was revised to ‘Previous studies indicated that plant adaptability was influenced by soil conditions and rhizosphere microbial taxa [55,56]’ (line 414, 415).
The partial content ‘In this study, apart from a significantly lower Shannon index of bacterial community observed in August for the introduced straight type compared to the straight type, no statistically significant differences were found in other indexes between the two types (Figure 2). These findings suggested that a similarity in soil microbial communities played a crucial role in the successful introduction of the straight type. Although the introduced straight type had the same origin region as the straight type, the introduced straight type exhibited significantly smaller DBH, BSW and BSR (p < 0.05) (Table 1)’ was revised to ‘(In this study, although the introduced straight type had the same origin region as the straight type, the introduced straight type exhibited significantly smaller DBH, BSW and BSR (p < 0.05) (Table 1))’ (lines 415 to 418).
The ‘indicated’ was revised to ‘indirectly suggested’ (line 420).
The partial content ‘Meanwhile, the prediction results of soil bacterial functions showed that nitrogen fixation abundance was significantly lower in the introduced straight type compared to the straight type at each stage (Figure 4A, Table S3), which may be an important factor leading to the low soil NH4+-N content of the introduced straight type [51]’ was revised to ‘Meanwhile, apart from a significantly higher Shannon index of bacterial community observed in August for the introduced straight type compared to the straight type, no statistically significant differences were found in other indexes between the two types (Figure 2). The prediction results of soil bacterial functions showed that nitrogen fixation abundance was significantly lower in the introduced straight type compared to the straight type at each stage (Figure 4A, Table S3), which may be an important factor leading to the low soil NH4+-N content of the introduced straight type [51]’ (lines 422 to 429).
The partial content ‘The plant-microbe interactions were likely to influence fitness of plant, as supported by previous research’ was revised to ‘The interaction among soil, plants and soil microbes was likely to influence adaptation of plant, as supported by previous research’ (line 436, 437).
The partial content ‘Soil microbes associated with rapid growth of D. sinicus’ was revised to ‘Potential key microbes associated with rapid growth of D. sinicus’ (line 450).
The partial content ‘Correlation analysis highlighted a positive association between soil NH4+-N content and bacterial genera Reyranella, Paenibacillus, Micromonospora, Nocardioides, Luedemannella, Pedomicrobium, Bradyrhizobium (p < 0.05), as well as fungal genera Trichoderma, Mortierella, Trechispora, Apiotrichum, Exophiala, Acremonium, Cordana. Among them, Bradyrhizobium and Paenibacillus from bacterial communities, and Trichoderma from fungal communities may contribute to increased NH4+-N availability for D. sinicus [15,17,64,65]. The enrichment of these microorganisms was a direct response to the increased demand for NH4+-N during rapid growth of D. sinicus’ was revised to ‘Correlation analysis highlighted a positive association between soil NH4+-N content and bacterial genera Reyranella, Paenibacillus, Micromonospora, Nocardioides, Luedemannella, Pedomicrobium, Bradyrhizobium, as well as fungal genera Trichoderma, Mortierella, Trechispora, Apiotrichum, Exophiala, Acremonium, Cordana (p < 0.05). Among them, the taxa associated with N fixation, such as Bradyrhizobium, Paenibacillus, and Trichoderma, may contribute to increased soil NH4+-N availability [12,14,64,65], which could facilitate the rapid growth of D. sinicus’ (lines 463 to 469).
In the materials and methods, the format of Table 2 was adjusted (line 481).
In the conclusions, the word ‘suggests’ was revised to ‘suggested’; The word ‘exhibits’ was revised to ‘exhibited’ (line 560).
The partial content ‘Bacterial genera Bradyrhizobium and Paenibacillus, as well as the fungal genus Trichoderma, may contribute to the improvement of NH4+-N content in the rhizosphere soil during the rapid growth period. Moreover, compared with the native habitat of the straight type, the soil microbial functions of the introduced straight type exhibited a significant decrease in nitrogen fixation, and dominant microbial function was associated with ureolysis. It may reflect that the soil properties have a great impact on rhizosphere microbial communities. Saprophytic fungi were the dominant fungal functional taxa in both straight and introduced straight type soils. Thus, the enrichment and functions of soil microbes play a crucial role in facilitating the rapid growth and adaptability of D. sinicus. when introducing and cultivating woody bamboo, it is imperative to not only select superior seed sources, but also consider the influence of microbes on the bamboo forest in order to enhance adaptation and growth’ was revised to ‘Moreover, compared with the native habitat of the straight type, the soil bacterial functions of the introduced straight type exhibited a significant decrease in nitrogen fixation, and dominant microbial function was associated with ureolysis. Saprophytic fungi were the dominant fungal functional taxa in both straight and introduced straight type soils. It may reflect that bacterial communities were more susceptible to soil conditions compared to fungal communities. This paper constitutes one of the initial researches exploring the effects of rhizosphere microorganisms in woody bamboo on its adaptability and rapid growth. When introducing and cultivating woody bamboo, it is imperative to not only select superior seed sources, but also consider the influence of microbes on the bamboo forest in order to enhance adaptation and growth’ (lines 561 to 571).
In the supplementary materials, the partial content ‘Abund_Jaccard’ was revised to ‘abund-jaccard’ (line 573).
In the references, the reference for [54] was revised from ‘Thiergart, T.; Durán, P.; Ellis, T.; Vannier, N.; Garrido-Oter, R.; Kemen, E.; Roux, F.; Alonso-Blanco, C.; Ågren, J.; Schulze-Lefert, P.; et al. Root microbiota assembly and adaptive differentiation among European Arabidopsis populations. Nat. Ecol. Evol. 2020, 4, 122-131. https://doi.org/10.1038/s41559-019-1063-3.’ to ‘Fang, X.Y.; Yu, T.B.; Yang, L.; Zang, H.D.; Zeng, Z.H; Yang, Y.D. Progress of bacterial wilt and its soil micro-ecological regulation in peanut. Chin. J. Eco-Agric. 2023. https://link.cnki.net/urlid/13.1432.S.20230815.1123.001.’ (line 729, 730).
The reference for [56] was revised from ‘Andrade, L.R.M.; Aquino, F.G.; Echevarria, G.; Oliveira, J.S.; Pereira, C.D.; Malaquias, J.V.; Souza, K.S.; Montargès-Pelletier, E.; Faleiro, F.G.; Junior, F.B.R.; et al. Edaphic factors as genetic selection agents and adaptation drivers of native plant species in harsh environments of the Brazilian savanna. Plant Soil. 2022, 479, 301-323. https://doi.org/10.1007/s11104-022-05520-3.’ to ‘Thiergart, T.; Durán, P.; Ellis, T.; Vannier, N.; Garrido-Oter, R.; Kemen, E.; Roux, F.; Alonso-Blanco, C.; Ågren, J.; Schulze-Lefert, P.; et al. Root microbiota assembly and adaptive differentiation among European Arabidopsis populations. Nat. Ecol. Evol. 2020, 4, 122-131. https://doi.org/10.1038/s41559-019-1063-3.’ (lines 734 to 736).

Round 2
Reviewer 2 Report
Authors revied the manuscript